# Biological Characterization and Whole-Genome Analysis of *Bacillus subtilis MG-1* Isolated from Mink Fecal Samples

**DOI:** 10.3390/microorganisms11122821

**Published:** 2023-11-21

**Authors:** Jianwei Ren, Detao Yu, Nianfeng Li, Shuo Liu, Hang Xu, Jiyuan Li, Fang He, Ling Zou, Zhi Cao, Jianxin Wen

**Affiliations:** College of Veterinary Medicine, Qingdao Agricultural University, Qingdao 266109, China; a17852022250@outlook.com (J.R.); a15725674736@outlook.com (D.Y.); l18464313601@163.com (N.L.); ls166411@163.com (S.L.); xuhangxhxhxh@163.com (H.X.); ljy13356752156@163.com (J.L.); hefang1030@126.com (F.H.); zouling2952@126.com (L.Z.); cz_ndnxy@126.com (Z.C.)

**Keywords:** *mink*, *Bacillus subtilis*, biological characteristics, whole genome, probiotics

## Abstract

*Bacillus subtilis* is an important part of the gut microbiota and a commonly used probiotic. In the present study, to assess the biological characteristics and probiotic properties of *B. subtilis* derived from mink, we isolated *B. subtilis MG-1* isolate from *mink* fecal samples, characterized its biological characteristics, optimized the hydrolysis of casein by its crude extract, and comprehensively analyzed its potential as a probiotic in combination with whole-genome sequencing. Biological characteristics indicate that, under low-pH conditions (pH 2), *B. subtilis MG-1* can still maintain a survival rate of 64.75%; under the conditions of intestinal fluid, gastric acid, and a temperature of 70 °C, the survival rate was increased by 3, 1.15 and 1.17 times compared with the control group, respectively. This shows that it can tolerate severe environments. The results of hydrolyzed casein in vitro showed that the crude bacterial extract of isolate *MG-1* exhibited casein hydrolyzing activity (21.56 U/mL); the enzyme activity increased to 32.04 U/mL under optimized reaction conditions. The complete genome sequencing of *B. subtilis MG-1* was performed using the PacBio third-generation sequencing platform. Gene annotation analysis results revealed that *B. subtilis MG-1* was enriched in several Kyoto Encyclopedia of Genes and Genomes (KEGG) metabolic pathways, and most genes were related to Brite hierarchy pathways (1485–35.31%) and metabolism pathways (1395–33.17%). The egg-NOG annotation revealed that most genes were related to energy production and conversion (185–4.10%), amino acid transport and metabolism (288–6.38%), carbohydrate transport and metabolism (269–5.96%), transcription (294–6.52%), and cell wall/membrane/envelope biogenesis (231–5.12%). Gene Ontology (GO) annotation elucidated that most genes were related to biological processes (8230–45.62%), cellular processes (3582–19.86%), and molecular processes (6228–34.52%). Moreover, the genome of *B. subtilis MG-1* was predicted to possess 77 transporter-related genes. This study demonstrates that *B. subtilis MG-1* has potential for use as a probiotic, and further studies should be performed to develop it as a probiotic additive in animal feed to promote animal health.

## 1. Introduction

The Mustelidae family is the largest family within the order Carnivora and includes *weasels*, *minks*, *polecats*, and *otters*. The gut microbiota of different members of the Mustelidae family has been investigated in previous studies [1]. However, the gut microbiota of *weasels* and *polecats* has not been adequately studied. Most studies on the gut microbiota of the Mustelidae family have focused on *minks* and *otters* [2,3]. The *mink* (Neovison vison, hereafter referred to as *mink*) is an economically important animal, with its fur having the main commercial value. At mink farms, the main diet is rich in proteins and contains feed additives to ensure the quick absorption of nutrients, improved growth performance, and the production of higher quality fur [4,5]. Moreover, *minks* possess a short gut, which leads to the quick digestion and absorption of food. This could be because of their unique gut microbiota. Previous studies have reported that the gut microbiota of minks is influenced by age and diet [6,7].

The gut microbiota is a large and complex group of bacteria [8,9]. It plays an important role in the host body, mainly in organizing and developing gut cells, maintaining the energy balance in the body, regulating gut pH, and promoting immune system development [10,11]. In 2014, the FAO/WHO defined probiotics as live micro-organisms that, when administered in adequate amounts, confer a health benefit on the host [12]. Probiotics must possess certain characteristics, such as tolerance to harsh growing environments, before they can be recognized and widely used [13]. The most commonly used probiotics are *Lactobacillus* spp., *Bifidobacterium* spp., *Lactococcus* spp., *Streptococcus* spp., *Enterococcus* spp., *Bacillus* spp., and *Saccharomyces cerevisiae* [14]. Among them, *Bacillus* spp. can produce spores that are highly tolerant to harsh environments, such as high and low temperatures and low pH. Moreover, the genus exhibits anti-cancer and antioxidant properties and produces antibacterial compounds and vitamins as secondary metabolites; therefore, several species of the genus are used as probiotics [15], and *Bacillus cereus* [16], *Bacillus clausii* [17], *Bacillus coagulans* [18], *Bacillus licheniformis* [19], *Bacillus polyfermenticus* [20], *Bacillus pumilus* [21], and *B. subtilis* [22] have been commercialized as such.

Casein is the main protein in cow, sheep, and human milk [23,24,25]. After hydrolysis, a variety of functional small peptides can be produced which can participate in immune regulation, neuroregulation, and intestinal barrier function regulation [26,27]. *B. subtilis* produces extracellular enzymes when grown in a suitable environment [28]. These include eight extracellular proteases [29]. At present, there are few studies on the in vitro hydrolysis of casein by *B. subtilis*, and whether its extracellular protease can hydrolyze casein in vitro needs further study.

*B. subtilis* is a widely studied Gram-positive bacterium. It can produce and secrete large quantities of enzymes [30]; therefore, it is used in molecular biology [31] and various industries, such as pharmaceutics and fermentation [32]. Moreover, *B. subtilis* can produce antibiotics such as amikomycin A and non-amikomycin that work against *Helicobacter pylori* infections [33]. Starosila [34] reported the antiviral activity of peptide P18 produced by *B. subtilis 3* against influenza virus and validated its potential as an antiviral compound. Some *B. subtilis* strains produce enterotoxins [35]. Therefore, safety assessments of *B. subtilis* strains, based on toxin production, antibiotic resistance, and biogenic amine production [15], are essential before they can be used as probiotics. With the development of high-throughput sequencing technology, whole-genome sequencing and analysis can be used to evaluate the safety of probiotics from the gene level, which greatly improves the efficiency of evaluation [36].

In addition to this, an important role of whole genome sequencing is to predict the function of bacteria; that is, the potential of bacteria to act as probiotics [37]. In this regard, whole-genome sequencing analysis shows the whole gene sequence of a bacterium and predicts its open reading frame, and compares this with KEGG, GO, egg-NOG, and other databases to annotate proteins in the bacterial genome and predict functional genes [38].

In the present study, we investigated the biological and genomic characteristics of *B. subtilis MG-1* isolated from fecal samples of *mink*. We explored the casein-hydrolyzing ability of its crude extract and optimized the reaction conditions. Moreover, the genomic characteristics of *B. subtilis MG-1* were comprehensively evaluated for its use as a potential probiotic.

## 2. Materials and Methods

### 2.1. Source, Isolation, and Purification of Isolates

Fifty fresh fecal samples were collected in 10 mL sterile centrifuge tubes from minks aged 90–120 days from a farm in Qingdao, China. The *minks* were fed a mixture of chicken rack, duck liver, cod, wheat flour, and vitamin supplements. The animal experiments performed in this study strictly followed the national guidelines for experimental animal welfare produced by the Ministry of Science and Technology of the People’s Republic of China in 2006 (Guiding Opinions on Kindly Treating Laboratory Animals. Relevant link: https://www.most.gov.cn/xxgk/xinxifenlei/fdzdgknr/fgzc/gfxwj/gfxwj2010before/201712/t20171222_137025.html (accessed on 13 May 2022)) and were approved by the Animal Welfare and Research Ethics Committee at Qingdao Agricultural University, Shandong, China (approval no: 2022-43). The fecal samples were transported at room temperature to the Laboratory of Veterinary Microbiology and Immunology, College of Animal Medicine, Qing-dao Agricultural University, and separated and purified. An amount of 10 g of fecal matter was taken for each sample and thoroughly mixed with 100 mL normal saline. It was then left to stand for 20 min, before taking 100 µL supernatant and evenly coating it on a Petri dish containing LB AGAR medium (Qingdao Haibo Biotechnology Co., Qingdao, China). According to the morphological characteristics of *B. subtilis*, the LB AGAR medium and the stripe plate method were used to select suspected *B. subtilis* isolates and further purify them. Isolate identification was performed using 16S rDNA after obtaining isolates.

### 2.2. Biological Characterization of the Isolated Isolate

#### 2.2.1. Growth Curve

A total of 100 µL *B. subtilis MG-1* bacterial liquid was inoculated in 9900 µL LB broth (Qingdao Haibo Biotechnology Co., Qingdao, China) and cultured at 37 °C 24 h for activation. Then, 1 mL of activated *B. subtilis MG-1* bacterial liquid (with a concentration of bacterial liquid of 2.6 × 10^9^ CFU/mL) was inoculated in 99 mL LB broth and cultured at 37 °C 24 h. Next, 200 µL of bacterial culture was added to a 96-well plate, and optical density at 600 nm (OD_600_) was measured using a microplate reader (Tecan Trading AG, Männedorf, Switzerland) at 0, 2, 4, 6, 8, 10, 12, 14, 16, 18, 20, 22, and 24 h. Parallel trials with three wells were set up for each time period. After the measurement was completed, a growth curve was drawn and the OD_600_ value of each time period was subtracted from the OD_600_ value at 0 h as the reference value.

#### 2.2.2. Tolerance of the Isolate

A pH tolerance test was carried out following the methodology of Kwaku and Kuebutornye et al. [39,40], with slight modification. The 1 mL of activated *B. subtilis MG-1* bacterial liquid was inoculated in 9 mL LB broth at different pH levels (pH 2, 3, or 4). The control group was inoculated in LB broth with normal pH. Each gradient was carried out 3 times in parallel. All samples were incubated at 37 °C and 220 rpm to assess the acid resistance ability of the isolate, and OD_600_ was measured at 0 and 24 h. The relative survival rates of the bacterial isolate were calculated using OD_600_ values.

Relative survival rates were calculated using the following formula:Relative survival rate (%) = (A_1_ − A_10_)/(A_2_ − A_20_) × 100%

**Remark 1.** 
*A_1_ represents the OD_600_ value at the end of the experimental group; A_10_ represents the OD_660_ value at 0 h of the experimental group; A_2_ represents the OD_600_ value at the end of the control group; A_20_ represents the OD_600_ value at 0 h of the control group.*


An intestinal fluid tolerance test was carried out following the methodology of Kwaku and Kuebutornye et al. [39,40], with slight modification. In the intestinal fluid tolerance test, the bacterium was inoculated in LB broth containing 0.03% bile salts and 1% trypsin, and OD_600_ was measured at 0 and 24 h. The control group was inoculated in LB broth with normal operation. Set three parallel sets. In addition, the bacterial culture was incubated in a constant temperature shaker incubator at 37 °C and 220 rpm. The relative survival rates of the bacterial isolate were calculated using OD_600_ values.

A stomach tolerance acid test was carried out following the methodology of Kwaku and Kuebutornye et al. [39,40], with slight modification. The stomach acid tolerance ability of the isolates was assessed, and 1 mL of bacterium was inoculated in 9 mL LB broth containing 1% pepsin (pH 3) (MD Bio, Inc., Qingdao, China). The bacterial culture for the control group was grown in LB broth without any treatment. Each gradient was carried out 3 times in parallel. And the OD_600_ was measured at 0, 0.5, 1, 1.5, 2, 2.5, and 3 h. The bacterial culture was incubated in a constant temperature-shaker incubator (Shanghai Minquan Instruments Co., Ltd., Shanghai, China) at 37 °C and 220 rpm. The relative survival rates of the bacterial isolate were calculated using OD_600_ values.

For the bile salt tolerance analysis, 1 mL *B. subtilis MG-1* bacterial liquid was cultured in 9 mL LB broth containing 0.15% bile salts (Shanghai Macklin Biochemical Co., Ltd., Shanghai, China). The bacterial culture for the control group was grown in LB broth without any treatment. Set three parallel sets. All samples were incubated at 37 °C and 220 rpm. Then, 100 μL of the aforementioned bacterium was inoculated in LB nutrient agar (Qingdao Haibo Biotechnology Co., Qingdao, China) at 0 h and 3 h. It was then cultured at 37 °C for 12 h, colony counting was performed, and the survival rate was calculated.

A heat tolerance test was carried out following the methodology of Kwaku and Kuebutornye et al. [39,40], with slight modification. For the heat tolerance test, 1 mL *B. subtilis MG-1* bacterial liquid was inoculated in 9 mL LB broth and incubated at 50 °C, 60 °C, and 70 °C for 10 min. The bacterial culture for the control group was grown in LB broth without any treatment. Each gradient was carried out 3 times in parallel. All were incubated at 37 °C and 220 rpm, the OD_600_ was measured at 0 and 24 h, and the survival rates were calculated.

### 2.3. Determination of Casein-Hydrolyzing Activity of Bacterial Extract

Different concentrations of a standard solution of L-tyrosine (Sinopharm Chemical Reagent Co., Ltd., Shanghai, China) were prepared: 5 mL of 0.4 mol/mL Na_2_CO_3_ solution (Tianjin basifu Chemical Trade Co., Ltd., Tianjin, China) and 1 mL of Folin–Ciocalteu reagent (Beijing Solarbio Science & Technology Co., Ltd., Beijing, China) were added to the standard solutions, the mixtures were placed in a water bath at 40 °C for 20 min, and absorbance was measured at 660 nm using a microplate reader. Next, a standard curve was prepared, and a regression equation was established.

We used the Folin phenol method to determine the ability of the metabolites of *B. subtilis MG-1* to hydrolyze proteins. Activated *B. subtilis* was inoculated in LB broth (10% culture) at 37 °C and 220 rpm and incubated for 24 h. The cultured bacterial solution was centrifuged at 10,000 rpm for 10 min to obtain the supernatant. The supernatant was filtered through a 0.22 μm filter to obtain the bacterial extract. In the experimental and control groups, we added 1 mL of crude enzyme extract and 1 mL of deionized water, respectively; three replicates were used for each group. The extract was heated at 37 °C for 2 min in a constant temperature water bath (Shanghai qixin scientific instrument Co., Ltd., Shanghai, China), and the reaction was induced by adding 1 mL of 2% casein (Sinopharm Chemical Reagent Co., Ltd., Shanghai, China) solution (preheated at 40 °C for 5 min) and incubating at 37 °C for 10 min. The reaction was terminated by adding 2 mL of 0.4 mol/L TCA (Tianjin basifu Chemical Trade Co., Ltd., Tianjin, China) solution, and the mixture was thoroughly mixed and left for 10 min.

After filtering the terminated reaction mixture using qualitative filter paper, 5 mL of 0.4 mol/L Na_2_CO_3_ solution and 1 mL of Folin-phenol reagent were added to 1 mL of the filtrate, and then it was mixed thoroughly and left to develop at 37 °C for 20 min in a constant temperature water bath. Next, we added deionized water to the reaction mixture, and the volume was fixed to 20 mL. Finally, we measured the absorbance at 660 nm using a UV spectrophotometer (Beijing General analytical Instrument Co., Ltd., Beijing, China), wherein the absorbance of the control group was set at 0.

Enzyme activity of the test group was measured using the following equation:Enzyme activity (U) = (D_1_ − D_2_) × K × V × N/T

**Remark 2.** 
*D_1_ represents the OD_660_ value of the experimental group; D_2_ represents the OD_660_ value of the control group; K represents the constant of the L-tyrosine standard curve; V represents the total reaction volume (mL); n represents the dilution ratio; and T represents the reaction time (min).*


### 2.4. Influence of Temperature and pH on Enzyme Activity in Crude Bacterial Extract

To determine the effect of pH on the enzymatic activity of the metabolites, we adjusted the pH of the 2% casein solution to 6, 7, 8, 9, and 10 and determined the enzyme activity. The optimal pH was selected based on the results, and the reaction was carried out under different reaction temperatures (30 °C, 40 °C, 50 °C, 60 °C, 70 °C, and 80 °C) to determine the enzyme activity of the metabolites. Next, the effect of temperature on the enzyme activity of the extract was determined. The crude enzyme extract was heated at 30 °C, 40 °C, 50 °C, 60 °C, 70 °C, and 80 °C for 30 min under an optimal pH and reaction temperature, and the enzyme activity was determined using the method described in Section 2.3.

### 2.5. Colony Preparation and DNA Extraction

After activation, 1 mL culture was inoculated in 99 mL LB broth and incubated at 37 °C and 220 rpm for 24 h. After 24 h, the bacterial culture was centrifuged at 9000 rpm at 4 °C for 8 min to remove the supernatant, and the pellet was washed and resuspended in an appropriate amount of 1 × PBS buffer (Shanghai Beyotime Biotechnology Co., Ltd., Shanghai, China). The resuspended bacterial pellet was centrifuged again at 9000 rpm and 4 °C for 8 min. The pellet was washed again 1–2 times until the supernatant was clear. The supernatant was removed, and the bacterial pellet was collected. The abovementioned procedure was repeated until the weight of the bacterial pellet was ≥0.5 g, and it was quickly frozen in liquid nitrogen for 15–20 min. The CTAB method was used to extract genomic DNA from the frozen pellet. The isolated DNA was quantified using a UV spectrophotometer and subjected to 1% agarose gel electrophoresis to determine its integrity.

### 2.6. Library Construction and Sequencing

We used the TruSeq Nano DNA LT Library Prep Kit (Illumina, San Diego, CA, USA) to construct the genomic library. Next, an MiSeq Reagent Kit V3 (600 cycles) was used to perform two-end sequencing of 2 × 300 bp using the Illumina NovaSeq and Pacbio Sequel platforms.

### 2.7. Quality Control and Sequence Assembly

We used the FastQC (v0.11.7) program to control the quality of the data, and Adapter Removal (v2.2.2) was used to remove the contamination of the 3’-end joint. Next, the quality of all reads was corrected using the SOAPec (v2.03) program, which corrects errors based on Kmer frequency, which was set to 17. To obtain contigs, we used the HGAP and CANU tools for de novo genome assembly using the PacBio reads, and pilon software (v1.18) was used to correct the high-quality data of the second generation of contigs. Finally, we used pilon software (v1.18) to assemble the third generation of contigs to obtain complete sequences.

### 2.8. Gene Prediction

We used the GeneMarkS program for gene prediction of the complete gene sequence. tRNA, rRNA, and sRNA were predicted in the genome using tRNA scan-SE, Barrnap (v0.9) software, and the Rfam database, respectively. CRISPRCs finder was used to predict highly conserved regions (DRs) (forward repeat sequences) and spacers (spacer regions) in the whole genome. The RECON (version 1.08), Repeat Scout (version 1.0.5), and NUCmer (NUCleotide MUMmer; version 3.1) software were used for repeat sequence analysis. PhiSpy, Island Viewer 4, BLAST, and hmmscan (v3.2.1) software were used to predict prophages, gene islands, and carbohydrate-active enzymes (CAZys) in the genome.

### 2.9. Functional Gene Annotation

The SignalP (v4.1) and TMHMM (v2.0) software were used to predict the signal peptide sequence and transmembrane helix structure in protein-coding genes, respectively; the proteins with signal peptide structure and without a transmembrane structure were screened as secreted proteins. Sequences were aligned in the NR and egg-NOG (COG) databases using diamond (v0.8.36) and diamond blastp software to complete the non-redundant (NR) protein annotation and eggNOG annotation of protein coding genes. KEGG Orthology (KO) and pathway annotation of protein-coding genes were performed using the KAAS (v2.1) automated annotation system of the Kyoto Encyclopedia of Genes and Genomes (KEGG) database. Diamond, BLAST2GO (v1.0) software, and TCDB (Transporter Classification Database) were used for Swiss-Prot, Gene Ontology (GO), TCDB functional, protein domain, and pathogen–host interaction annotations of the protein coding genes, respectively.

## 3. Results

### 3.1. Morphology of B. subtilis MG-1 Isolate

The bacterial colony characteristics of B. subtilis were used for isolation and purification, and a target isolate was obtained (Figure 1). The isolate was identified as *B. subtilis* and named *MG-1*.

### 3.2. Growth Curve Analysis of B. subtilis MG-1

Growth curve analysis revealed that the growth of the *MG-1* isolate was slow from 0 h to 14 h, suggesting that this was the adjustment phase; however, the growth rapidly increased from 14 h to 20 h, and the increase was the fastest from 14 h to 16 h, indicating that this was the logarithmic growth phase. Between 20 and 24 h, the growth stabilized, and the bacterium entered the stationary phase (Figure 2a).

### 3.3. Tolerance of B. subtilis MG-1 to Various Harsh Environments

The acid tolerance test revealed that at pH 2, *B. subtilis MG-1* exhibited the lowest relative survival rate of 64.75%; at pH 3 and 4, the relative survival rates were 75.95% and 77.73%, respectively (Figure 2b).

The intestinal fluid tolerance test revealed that the relative survival rate of *B. subtilis MG-1* was three times higher than in the control group under the simulated intestinal fluid environment (Figure 2c). The gastric acid tolerance test elucidated that *B. subtilis MG-1* did not exhibit any growth in the experimental group over 0.5–1 h, whereas its growth in the control group was normal. At a bile salt concentration of 0.15%, the survival rate of isolate *MG-1* was 28.9%.

However, after 1 h, the experimental group exhibited growth and the relative survival rate was 1.15 times higher than in the control group, thus suggesting a stronger tolerance to the gastric acid environment (Figure 3a). In addition, the heat tolerance analysis elucidated that the relative survival rates of *B. subtilis MG-1* at 50 °C, 60 °C, and 70 °C were 2.36, 0.81, and 1.17 times higher than in the control group, respectively (Figure 3b).

### 3.4. Enzyme Activity of B. subtilis MG-1 Metabolites

The casein-hydrolyzing activity of the *B. subtilis MG-1* extract was 21.56 U/mL under normal conditions, as per the standard curve constructed using L-tyrosine (Figure 4a). We observed that the casein-hydrolyzing activity of the *B. subtilis* extract was higher than the initial activity at pH 6, 8, 9, and 10; the highest enzyme activity was 28.97 U/mL at pH 7 (Figure 4b). Moreover, with an increase in reaction temperature, the casein-hydrolyzing activity of the *B. subtilis* crude extract increased and was the highest (25.98 U/mL) at 50 °C (Figure 4c). The enzyme activity gradually decreased from 50 °C to 80 °C and was the lowest (21.66 U/mL) at 80 °C. The absolute value of the slope at 50–70 °C was larger than that at 30–50 °C, indicating that the enzyme activity of the extract was highly affected by the reaction temperature above 50 °C; therefore, 50 °C was considered the optimal reaction temperature. Furthermore, we observed that the enzyme activity gradually decreased with an increase in temperature, and the highest activity (32.04 U/mL) was observed at 30 °C, whereas the lowest activity (22.79 U/mL) was observed at 80 °C (Figure 4d). The absolute value of the slope between 30 and 50 °C was larger than that between 50 and 80 °C, indicating that the stability of the extract was lower at >50 °C than between 30 °C and 50 °C. Therefore, the extract exhibited good enzyme activity and thermal stability at 30–80 °C.

### 3.5. Analysis of B. subtilis MG-1 Sequencing Data

We obtained a total of 8,461,188 reads containing 1,277,639,388 bases and 43.76% GC content through next-generation sequencing. Statistical analysis of the three generations of sequencing data resulted in a total of 1,274,433 sequences, with a genome length of 12,203,635,059 bp; the longest and shortest sequence lengths were 337,816 bp and 51 bp, respectively. The lengths of N20, N50, and N90 (where all sequence lengths were arranged from long to short, sequentially added in that order, and the length of the last sequence was calculated when the length of the addition reached 20, 50 and 90% of the total length of the sequence) were 12,772 bp, 9680 bp, and 7335 bp, respectively, and the GC content was 43.06%.

### 3.6. Analysis of Basic Genomic Information

The complete sequence information results obtained after quality control and gene splicing are shown in Table 1. *B. subtilis MG-1* possesses one circular chromosome (Figure 5). The analyses revealed that it did not contain any plasmid sequences.

### 3.7. Coding Genes and Non-Coding RNAs (ncRNAs)

Bioinformatics analyses of the sequencing data predicted a total of 4509 open reading frames (ORFs) as protein coding genes, accounting for 88.06% of the total genome length. The density of the ORFs was 1.035 genes per kb; the longest sequence length was 16,488 bp; and the average length was 850.85 bp. The length of the gene interval sequence was 520,377 bp, accounting for 11.94% of the genome sequence. The average GC content of the ORFs and gene interval was 44.17% and 38.29%, respectively.

Based on bioinformatics analyses, the following ncRNAs were predicted: 86 tRNAs, 30 rRNAs (10 5S rRNAs, 10 16S rRNAs, and 10 23S rRNAs), and 99 ncRNAs (remaining after the rRNA and tRNA were removed). The average tRNA sequence length was 77 bp, accounting for 0.1526% of the total genome sequence length. The average lengths of the three rRNA sequences were 111 bp, 1547 bp, and 2924 bp, respectively, accounting for 1.0519% of the total genome sequence length. The average length of ncRNA sequences was 153, accounting for 0.3491% of the total genome sequence length.

### 3.8. Prediction of Clustered Regularly Interspaced Short Palindromic Repeats (CRISPRs) in the Genome of B. subtilis MG-1

Two CRISPRs structures were predicted in the genome of *B. subtilis MG-1* (Table 2). Their lengths were 106 bp and 108 bp, and both had one spacer.

### 3.9. Prediction of Prophages, Genomic Islands, and CAZys in the Genome of B. subtilis

Nine prophages were predicted in the genome of *B. subtilis MG-1*, and their lengths were 13,049, 127,819, 30,211, 15,611, 103,555, 31,332, 21,041, 31, 154, and 7300 bp. The total length and the average length were 381,072 and 42,341, respectively.

Thirty-nine genomic islands were predicted in the genome of *B. subtilis MG-1* (Figure 6). Their lengths were as follows: 4732, 9161, 36,897, 8107, 4781, 4381, 8706, 7227, 4858, 5422, 4488, 5933, 74,373, 168,876, 18,736, 9446, 4925, 4166, 4907, 6819, 4584, 88,317, 16,475, 6805, 6644, 5413, 15,848, 5413, 9937, 4499, 29,523, 11,318, 6256, 21,994, 5518, 6658, 4075, 4087, and 13,665 bp. Moreover, the total length and average length of the genome islands were 663,970 and 17,024, respectively.

The CAZy prediction results for *B. subtilis MG-1* are shown in Figure 7. Six functions were predicted, including glycosyltransferase (GT), polysaccharide lyase (PL), carbohydrate esterase (CE), accessory activity (AA), carbohydrate binding module (CBM), and glycoside hydrolase (GH), with their numbers of annotated ORFs being 44, 7, 29, 7, 22, and 66, respectively, accounting for 0.0098%, 0.0016%, 0.0064%, 0.116%, 0.0049%, and 0.0146% of the total number of predicted ORFs, respectively.

### 3.10. Functional Annotation Analysis of Protein-Coding Genes

#### 3.10.1. egg-NOG Annotation Results

The egg-NOG annotation of the genome of *B. subtilis MG-1* revealed (Figure 8) that most ORFs were related to energy production and transformation (185–4.10%), chromosome division (31–0.68%), amino acid transport and metabolism (288–6.38%), nucleotide transport and metabolism (89–1.97%), carbohydrate transport and metabolism (269–5.96%), coenzyme transport and metabolism (116–2.57%), lipid transport and metabolism (87–1.92%), ribosome structure and biogenesis (164–3.63%), transcription (294–6.52%), recombination and repair (168–3.72%), cell wall/membrane/envelope biogenesis (231–5.12%), cell viability (37–0.82%), gene expression (3.72%), post-translational modification (111–2.46%), transport and metabolism of inorganic ions (206–4.56%), biosynthesis, transport, and catabolism of secondary metabolites (68–1.50%), signal transduction mechanism (152–3.37%), secretion and vesicle transport (32–0.70%), defense mechanism (63–1.39%), and fine extracellular structures (1–0.02%). Moreover, 1149 ORFs with unknown function comprised 25.48% of the total ORFs.

#### 3.10.2. KEGG Pathway Enrichment Results

The KEGG pathway enrichment analysis revealed that the genes of *B. subtilis MG-1* were enriched in eight types of functions (Figure 9). We observed that 1485 genes were enriched in the Brite hierarchies pathways, and were mainly involved in protein families’ signaling and cellular processes, genetic information processing, and metabolism; 168 genes were enriched in the cellular processes pathways, and were mainly related to cellular community prokaryotes, cell motility, cell growth and death, and transport and catabolism; 1395 genes were enriched in the metabolism pathways, and were mainly related to carbohydrate metabolism, amino acid metabolism, the metabolism of cofactors and vitamins, energy metabolism, nucleotide metabolism, lipid metabolism, the metabolism of other amino acids, the biosynthesis of other secondary metabolites, the metabolism of terpenoids and polyketides, xenobiotics biodegradation and metabolism, and glycan biosynthesis and metabolism; 314 genes were enriched in the environmental information processing pathways, and were mainly related to membrane transport and signal transduction; 220 genes were enriched in the genetic information processing pathways, and were mainly related to translation, replication and repair, folding, sorting, and degradation; 66 genes were enriched in the organismal systems pathways, and were mainly related to the endocrine system, aging, environmental adaptation, the nervous system, the immune system, the digestive system, and the excretory system; and 458 genes could not be annotated.

#### 3.10.3. GO Enrichment Results

GO annotation analysis revealed that gene functions could be divided into all three categories: biological process, cellular component, and molecular function (Figure 10). The analysis revealed that 8230 genes were enriched in biological process terms related to biological process, biosynthetic processes, cellular nitrogen compound metabolism, small-molecule metabolism, transport, catabolism, carbohydrate metabolism, protein maturation, cofactor metabolism, response to stress, lipid metabolism, cellular protein modification, translation, and signal transduction; 3582 genes were enriched in cellular process terms related to cells, cellular components, intracellular, built-in, plasma membrane, and organelle; and 6228 genes were enriched in molecular process terms related to molecular function, ion binding, DNA binding, oxidoreductase activity, lyase activity, nucleic acid binding transcription factor activity, kinase activity, isomerase activity, transferase activity, transferring acyl groups, RNA binding, and ligase activity.

#### 3.10.4. TCDB Annotation Results

TCDB annotation analysis revealed that 77, 259, 299, 49, 8, 27, and 126 genes were related to channels, electrochemical-potential-driven transporters, primary active transporters, group translocators, transmembrane electron carriers, accessory factors involved in transport, and incompletely characterized transport systems, respectively (Figure 11).

## 4. Discussion

Previous studies have associated numerous health benefits with specific microbes [41]. These include probiotic strains such as *B. subtilis MB40*, *Lactobacillus rhamnosus GG*, and *Bifidobacterium animalis subspecies lactis BB-12* [42]. The use of probiotics has rapidly increased over the past few years.

*B. subtilis* is widely used as a probiotic due to its beneficial effects on the body [43]. At the same time, because it can form spores under environmental stress conditions, preparations containing *B. subtilis* are more stable than those of other strains, and they are convenient to store [44]. In addition, the powerful function of *B. subtilis* also conditions it an advantage as a probiotic candidate [8]. Therefore, the comprehensive evaluation and verification of *B. subtilis* is essential.

In the present study, *B. subtilis MG-1* isolated from mink fecal samples could grow normally and reached the stationary phase at 20 h under normal culture conditions (constant temperature: 37 °C). This is consistent with the general growth law of bacteria. At the same time, this result exposes a problem, which may be caused by various factors (such as the source, culture environment, and geographical location) that influence the time point of each stage of growth of the isolates [45].

The biological characteristics of the isolate indicate its resistance to harsh environments. In the present study, the relative survival rate of *B. subtilis MG-1* was maintained at 64.75% at pH 2. This shows that it can maintain growth in a low-pH environment. A previous study by Toutiaee et al. showed that *B. subtilis* strains isolated from the digestive tracts of bees exhibited ≥100% survival rates at pH 4.0 and 6.0 [46]. This shows that *B. subtilis* can adapt to acidic environments. Combined with the results of this study, the growth ability of *B. subtilis* decreased with the decrease in pH, and in this process spores may be formed to resist changes in the external environment [45]. This requires further research.

Since probiotics are usually administered orally, they reach the stomach first [47]. The acidic conditions of the stomach and the presence of pepsin are the first hurdle for probiotics. In addition, other digestive enzymes also have an impact on the activity of probiotics [46]. In the present study, *B. subtilis MG-1* exhibited a high tolerance to simulated gastric acid conditions and maintained a high survival rate under 0.3% bile salt treatment. In a previous study, Tenea et al. [48] isolated a *B. subtilis* strain from bromeliads and reported that it was highly tolerant to synthetic gastric acid and 0.3% bile salts. This is similar to the results of this study.

The in vitro simulation of intestinal fluid showed some effect of intestinal fluid on bacterial growth [49]. In the present study, bacterial growth was enhanced under simulated intestinal fluid conditions, which showed that these intestinal fluid conditions promoted its growth. In previous studies, *B. subtilis* incubated with intestinal fluid showed higher survival rates [50]. But there is no evidence that it is this ingredient that promotes growth. In this study, the high survival rate shown by *B. subtilis* in simulated intestinal fluid was suspected to be related to the addition of trypsin. In addition, the isolate was isolated from *mink* feces, which may also be one of the reasons for its high tolerance to intestinal fluid.

In this study, *B. subtilis MG-1* showed high-temperature tolerance. This may be because the spores produced during high-temperature treatment resist the high temperature of the environment and begin to recover and grow after the temperature recovers. In a previous study, Kim et al. [51] reported that *B. subtilis* exhibited tolerance to a wide range of temperatures, i.e., 30 °C to 100 °C, which is consistent with the results of the present study.

Thus, *B. subtilis MG-1* exhibited tolerance to several extreme conditions, in which the physiological conditions of the host or gut environment were simulated, such as high fever, gastric acid, bile salt, strong acid, and intestinal juice. Therefore, its potential as a probiotic should be investigated in future studies.

In the present study, crude extracts of *B. subtilis MG-1* were able to hydrolyze casein, and their hydrolytic capacity improved under optimized conditions. Therefore, this isolate may help in the digestion and absorption of macromolecular proteins that cannot be digested by the human body. In previous studies, metabolites of some *B. subtilis* strains were reported to modulate the gut microbiota to reduce weight gain [52]. In minks, the quick digestion and absorption of a high-protein diet may be related to their intestinal flora. *B. subtilis* can hydrolyze proteins in vitro. A previous study reported that *B. subtilis* could hydrolyze feathers composed of natural and non-digestible proteins [53]. There are few reports on the hydrolysis of casein using crude extracts of *B. subtilis*, and the specific substances that produce hydrolysis were not purified in this study. The readily accepted explanation is that, during its growth, *B. subtilis MG-1* produces extracellular proteases that hydrolyze casein. However, Bacillus subtilis is known to produce many extracellular enzymes and other metabolites. Therefore, the mechanism of casein hydrolysis needs further study.

In addition to its biological characteristics, we characterized the genome of *B. subtilis MG-1* to further evaluate its potential as a probiotic. Various annotations were performed in this study, including CAZy annotation.

In the present study, genes encoding glycoside hydrolases were annotated in the genome of *B. subtilis MG-1*. In addition, most of the genes were related to glycosyl transferase, carbohydrate esterase, carbohydrate-binding module, and glucoside hydrolase. CAZys mainly include glycoside hydrolases, glycosyltransferases, polysaccharide lyases, carbohydrate esterases, and carbohydrate-binding module family enzymes [54]. Glycoside hydrolases exhibit functional diversity and are widely distributed in organisms [55]; they belong to the catabolic enzymes of carbohydrate metabolism pathways, including glycosidases, which hydrolyze glycosidic bonds, and glycosidases, which use disaccharides, oligosaccharides, or polysaccharides as sugar donors to catalyze the transfer of sugar residues to specific receptor molecules [56]. Glycosyltransferases can catalyze glycosidic bond formation using sugar donors containing nucleoside phosphate or lipid phosphate leaving groups [57]. Carbohydrate esterases can promote the hydrolysis of complex polysaccharides by glycosyl hydrolases [58]. Polysaccharide lyases depolymerize anionic polysaccharides containing uronic acids via a β-elimination mechanism [59]. In the present study, it was indicated that *B. subtilis MG-1* has a strong capacity for carbohydrate metabolism. Moreover, pathway enrichment analysis revealed that several *B. subtilis MG-1* genes were enriched in metabolic pathways, especially those related to carbohydrate metabolism. In addition, the strong tolerance of this isolate enabled it to still play an effective role in synthesizing and metabolizing carbohydrates in vivo.

In this study, we elucidated that the genome of *B. subtilis MG-1* possessed several genes related to amino acid metabolism, suggesting that it could actively participate in the synthesis and decomposition of amino acids in the intestine. In metabolic pathways, amino acids, the basic building blocks of proteins, play an important role in energy conversion and protein synthesis [60]. For example, glutamine is metabolized by glutamine-metabolizing enzymes (glutamine synthetase or glutamate dehydrogenase) to produce energy, and arginine is metabolized by nitric oxide synthase and arginase to produce nitric acid and polyamines, respectively [61]. A previous study reported that tryptophan degradation modulates kynurenine immune responses during infection, inflammation, and pregnancy; moreover, tryptophan could synthesize serotonin, and its availability affected susceptibility to mood disorders [62]. Several studies have reported that amino acid metabolism is closely related to the gut microbiota. For example, the three major tryptophan metabolic pathways of serotonin, kynurenine, and indole derivatives are directly or indirectly controlled by the gut microbiota. D-glutamate metabolized by the gut microbiota, as observed in patients with Alzheimer’s diseases, may affect glutamate N-methyl-D-aspartate receptors (NMDARs) and cognitive function in dementia patients [63].

In this study, we elucidated that the genome of *B. subtilis MG-1* possessed genes related to lipid metabolism, indicating its involvement in lipid metabolism in the body. Lipid metabolism includes the biosynthesis and degradation of lipids, such as fatty acids, triglycerides, and cholesterol. The gut microflora influences lipid metabolism and lipid levels in blood and tissues of mice and humans [64].

In the present study, the genome of *B. subtilis MG-1* possessed several genes involved in biosynthesis, which is in accordance with the results of previous studies. *B. subtilis* strains exhibit remarkable biosynthetic ability. The bioactive metabolites produced by *B. subtilis* mainly include non-ribosomal peptides (NRPs), polyketides (PKs), ribosomal peptides (RPs), and hybrid and volatile metabolites [65].

Therefore, this study elucidated that the *B. subtilis MG-1* isolate isolated from mink fecal samples exhibited characteristics similar to those of other probiotic strains belonging to the Bacillus genus; it exhibited tolerance to and growth in harsh environments. Moreover, its metabolites could hydrolyze casein in vitro. After optimizing the reaction conditions, its protein hydrolyzing ability was enhanced. Genomic characterization revealed that the genes of *B. subtilis MG-1* were involved in amino acid metabolism, lipid metabolism, and secondary metabolites’ synthesis, which further confirmed the probiotic properties of the strain.

## 5. Conclusions

In this study, *B. subtilis MG-1* was isolated from mink feces, and its biological characteristics, ability to hydrolyze casein, and whole-genome sequencing analysis showed that the isolate had good tolerance to environmental stress. The crude extract can hydrolyze casein in vitro. Genomic characterization demonstrates its potential as an effective probiotic. Future studies should investigate its use in feed additives and foods and the potential mechanisms of its various activities.

## Figures and Tables

**Figure 1 microorganisms-11-02821-f001:**
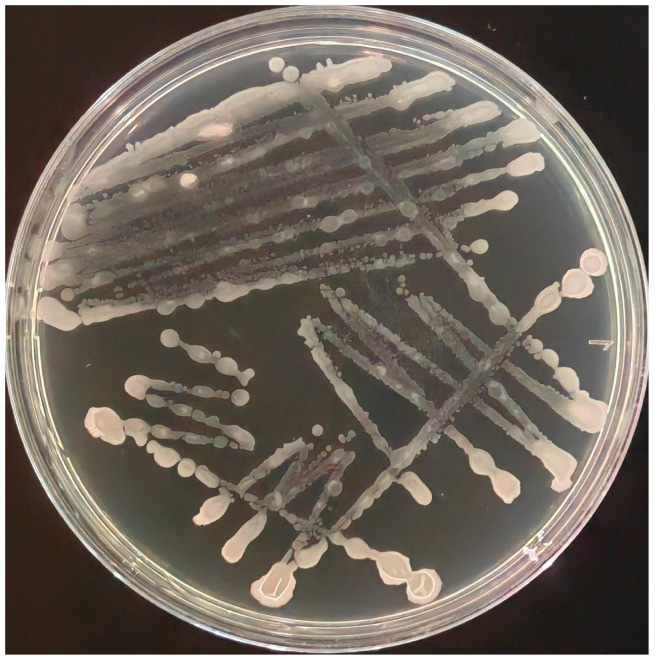
Colony morphology of the Bacillus subtilis isolate.

**Figure 2 microorganisms-11-02821-f002:**
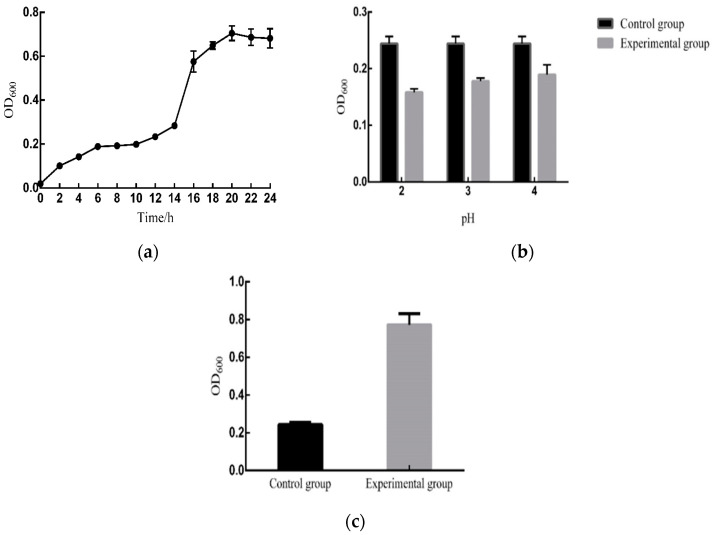
(**a**) Growth curve of *B. subtilis MG-1*. The abscissa is the incubation time, and the ordinate is the OD_600_ value. (**b**) Acid tolerance analysis of *B. subtilis MG-1*. The abscissa is three different gradients of pH 2, 3, and 4, and the ordinate is the OD_600_ value. (**c**) Intestinal fluid tolerance analysis of *B. subtilis MG-1*. The abscissa is the control group (black) and the experimental group (gray), and the ordinate is the OD_600_ value. In all figures, black represents the control group, and gray represents the experimental group.

**Figure 3 microorganisms-11-02821-f003:**
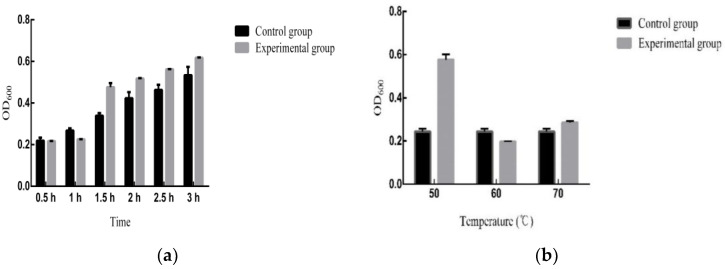
(**a**) Gastric acid tolerance analysis of *B. subtilis MG-1*. The abscissa is the incubation time, and the ordinate is the OD_600_ value. (**b**) Heat tolerance analysis of *B. subtilis MG-1*. The abscissa is three different temperature gradients of 50, 60, and 70 °C, and the ordinate is the OD_600_ value. In all figures, black represents the control group and gray represents the experimental group.

**Figure 4 microorganisms-11-02821-f004:**
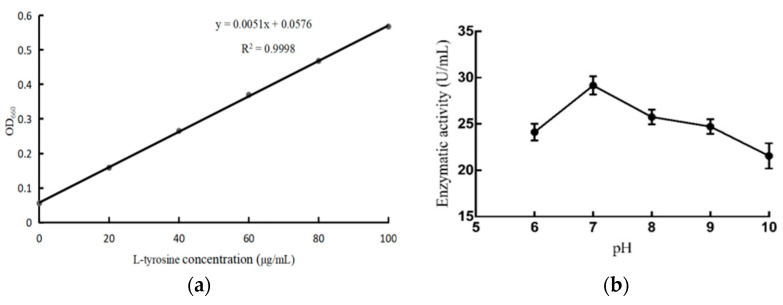
(**a**) Standard curve of L-tyrosine. The abscissa is the tyrosine concentration, and the ordinate is the OD_600_ value. (**b**) Effect of pH on enzyme activity of *B. subtilis MG-1* extract. The abscissa is different gradients of pH, and the ordinate is the enzyme activity. (**c**) Effect of reaction temperature on enzyme activity of *B. subtilis MG-1* extract. The abscissa is different gradients of reaction temperature, and the ordinate is the enzyme activity. (**d**) Effect of temperature on enzyme activity of *B. subtilis MG-1*. The abscissa is different gradients of temperature, and the ordinate is the enzyme activity.

**Figure 5 microorganisms-11-02821-f005:**
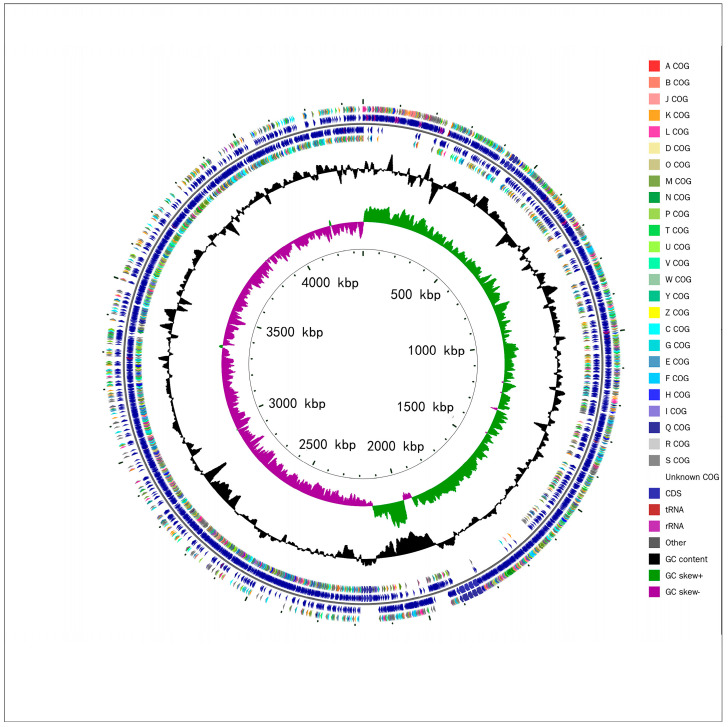
Chromosome of *B. subtilis MG-1*. From the inside out, the first circle represents the scale. The second circle represents GC Skew. The third circle represents GC content. The fourth and seventh circles represent the COG to which each CDS belongs. The fifth and sixth circles represent the positions of CDS, tRNA, and rRNA on the genome.

**Figure 6 microorganisms-11-02821-f006:**
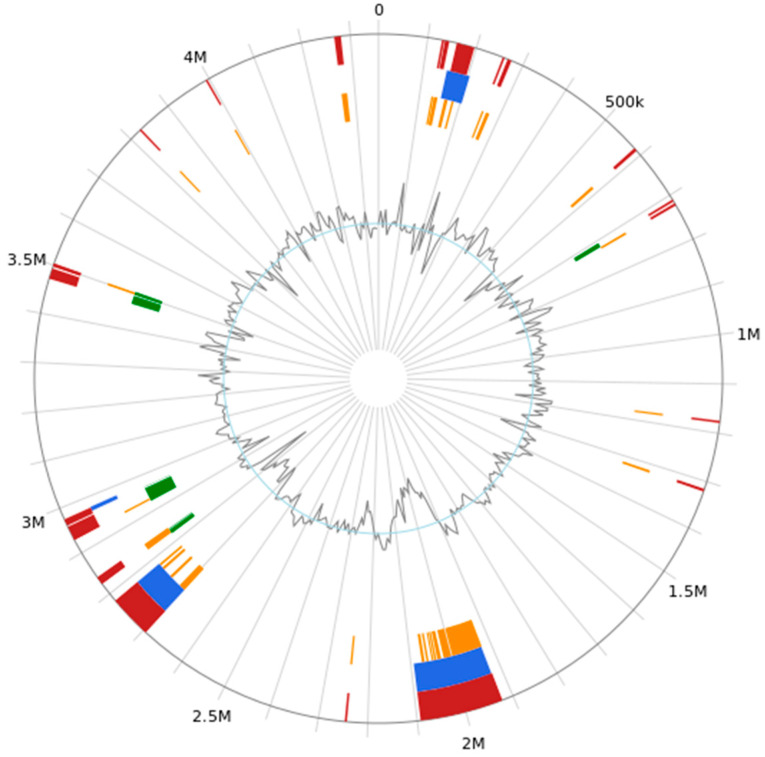
The genomic island predictions for *B. subtilis MG-1*. The red color represents the genomic island prediction results after the integration of the three methods (predicted with at least one method). Blue represents the prediction result of Island Path-DIMOB. Yellow represents the prediction results of SIGI-HMM. Green represents the prediction result of Island Pick. Light blue represents the Islander predictions.

**Figure 7 microorganisms-11-02821-f007:**
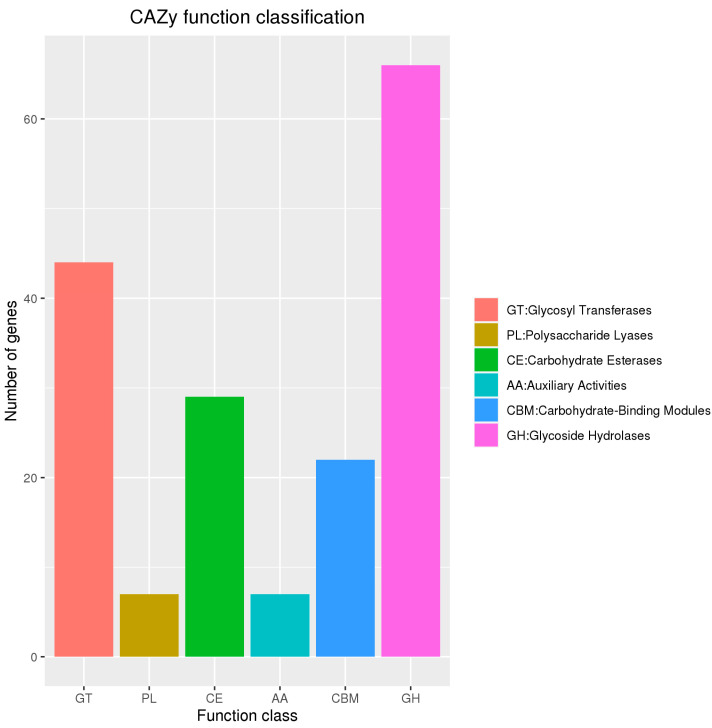
Prediction of carbohydrate-active enzymes (CAZys) in *B. subtilis MG-1* genome. The abscissa is the function class, and the ordinate is the number of matched genes.

**Figure 8 microorganisms-11-02821-f008:**
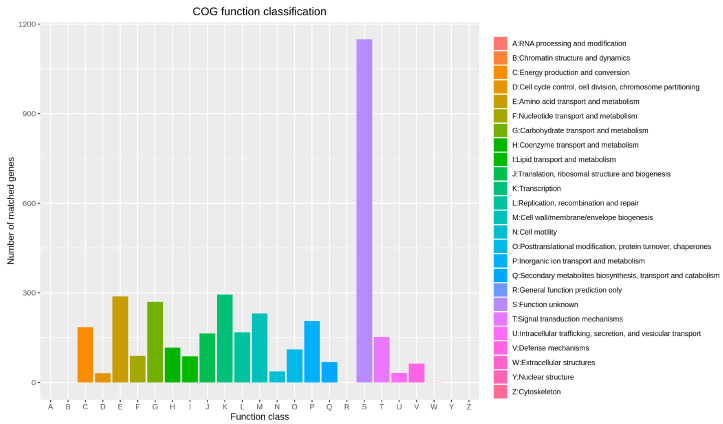
egg-NOG annotation of genome of *B. subtilis MG-1*. Different letters on the abscissa represent different function classes, and the ordinate is the number of matched genes.

**Figure 9 microorganisms-11-02821-f009:**
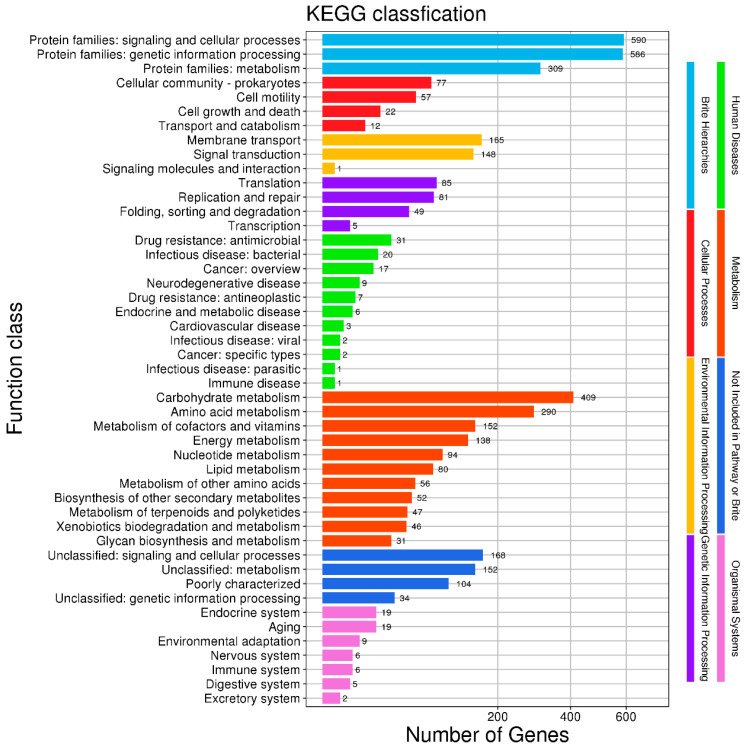
Kyoto Encyclopedia for Genes and Genomes (KEGG) pathway enrichment analysis of the genome of *B. subtilis MG-1*. The abscissa is the number of matched genes, and the ordinate is the function class.

**Figure 10 microorganisms-11-02821-f010:**
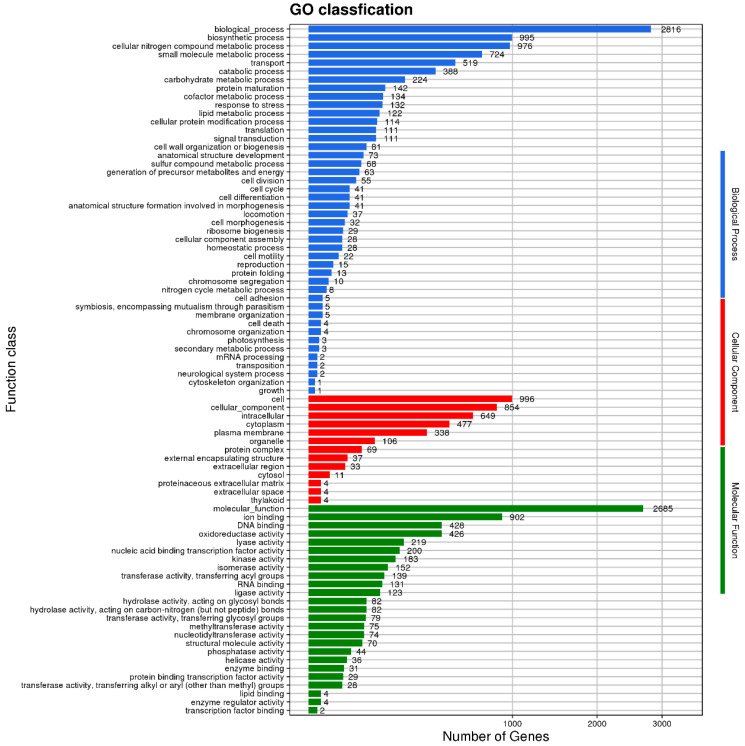
Gene Ontology (GO) annotation of the genome of *B. subtilis MG-1*. The abscissa is the number of matched genes, and the ordinate is the function class.

**Figure 11 microorganisms-11-02821-f011:**
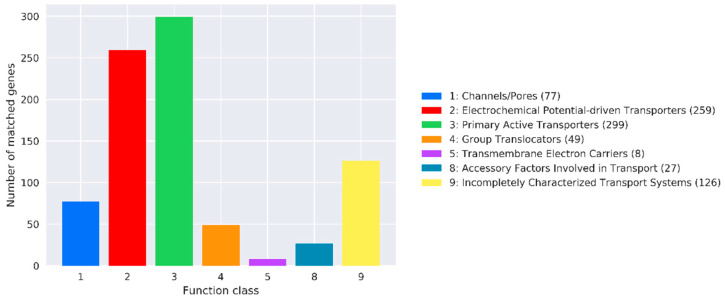
Transporter Classification Database (TCDB) annotation of the *B. subtilis* genome of *MG-1*. The abscissa represents TCDB first-order classification type, and the ordinate represents the number of genes on the annotation.

**Table 1 microorganisms-11-02821-t001:** Complete sequence information.

Seq Length (bp)	GC Content (%)	Seq Type
4,356,867	43.46	circular

In the table, “Seq Length” represents the total sequence length, “GC Content” represents the content of GC, and “Seq Type” represents the sequence type of the genome.

**Table 2 microorganisms-11-02821-t002:** Structure of *Bacillus subtilis MG-1* CRISPRs.

Start	End	Number of Spacers	Length (bp)
987,270	987,375	1	106
3,283,130	3,283,237	1	108

In the table, “Start” and “End” represent the start and end positions of the CRISPRs, respectively, “Number of Spacers” represents the number of CRISPRs spacers, and “Length” represents the length of CRISPRs.

## Data Availability

The data presented in the study are deposited in the NCBI repository, accession number CP110634. They can be found below: https://www.ncbi.nlm.nih.gov/nuccore/CP110634 (accessed on 15 May 2023).

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
