# Peer review of "Biological Characterization and Whole-Genome Analysis of Bacillus subtilis MG-1 Isolated from Mink Fecal Samples"

_microorganisms, 2023, doi:10.3390/microorganisms11122821_

Round 1

Reviewer 1 Report

Comments and Suggestions for Authors

Review of the article "microorganisms-2676699".

The article " Biological characterization and whole genome analysis of Bacillus subtilis MG-1 isolated from mink fecal samples" by Jianwei Ren, Detao Yu, Nianfeng Li, Shuo Liu, Hang Xu, Jiyuan Li, Fang He, Ling Zou, Zhi Cao and Jianxin Wen describes the biological and genetic characteristics of the strain. The authors sequenced the genomic DNA of Bacillus subtilis MG-1 on the PacBio platform, then, after ring chromosome assembly, analysed the genome of the strain and inferred the major functional characteristics of the identified coding sequences (CDS) using various databases. The authors also investigated some biological properties of MG-1. The purpose of the paper is not stated, did the authors just want to study the properties of the strain? The purpose should be stated.

Significant English language editing is required.

Notes:

Line 18-19 - " The complete genome analysis of B. subtilis MG-1 was performed 18 using the PacBio third-generation sequencing platform.". - The authors performed sequencing on this platform, not genome analysis.

Line 54 - Saccharomyces cerevisiae. - Should be italicised and the period before the reference should be deleted.

Lines 76-81 - The authors isolated only one strain on an agar plate? Such transport and storage results in losses of live microbial cells, so it is not surprising that a single strain was obtained. Did the authors inoculate before transport and freezing? Did they compare the results in terms of the number of cultured microorganisms before and after freezing? The authors should explain the purpose of their pre-freezing.

Also, on line 78 - according to the methodology described ...and separated and purified. It is not clear how they were separated and purified. This begs the question: was only one culture isolated? And how was it determined that it was the right strain? By what characteristics? Not by comparison with other bacteria in the gut microbiome.

Lines 80-81 - Biological characterisation of the isolated strain. - Should be moved to a new line and subheadings should be made for the biological characteristics listed below.

Probably in method 2.2. Growth curve line 82... the data on inoculum volumes and main experiment are mixed up. The inoculum should be in a smaller volume of medium.

Line 91 -1 ml B. Subtilis...it is not clear 1 ml of what?

The article nowhere mentions the number of microbes to be introduced.

Line 98 – «The relative survival rates of the bacterial strain were calculated using OD600 values» - it is not clear how the survival rate was calculated.

Line 103 – «And calculate the survival rate» - there is no indication of how this rate was calculated.

Line 145 – « ...2.5 Factors influencing enzyme activity in crude bacterial extract «- I would replace with «Influence of temperature and pH on enzyme activity in crude bacterial extract». 

Line 170 – «Illumina NovaSeq/Pacbio Sequel platform» - which platform did the authors use? If they used both platforms, that's how you should write it, with an "and".

In the Results section, there are no results for strain isolation, screening and purification.

Line 206 – « ... suggesting that this was the lag phase..»  Figure 1 (a) - how can the authors explain the increase in biomass between 0-14 hours if the authors refer to this period as the lag phase? It is well known that the lag phase is characterised by a lack of growth and sometimes even death of some of the cells in the system under the influence of environmental factors. It is likely that the growth curve of the strain studied on rich medium is characterised by a very short lag phase, which the authors failed to capture. The zero point also raises the question of why the OD is zero if the authors introduced some kind of inoculum (of unknown abundance).

It also raises the question of the need to present data on the growth curve of the strain on rich medium in the paper.

Line 216 - the authors should give the value for "The intestinal fluid tolerance test" not 316.26%, but e.g. "three times higher than in the control group". The same applies to lines 231-234.

Table 1 - why present it when the information is in the text?

In the caption to Figure 6, why decipher the colour scheme again when the figure itself provides the deciphering? The same applies to figures 8, 9 and 10.

Conclusion: The article needs significant revision before it can be decided whether it is suitable for publication.

Comments on the Quality of English Language

Extensive editing of English language required

Reviewer 2 Report

Comments and Suggestions for Authors

The manuscript submitted to “Microorganisms” an MDPI journal entitled: “Biological characterization and whole genome analysis of Bacillus subtilis MG-1 isolated from mink fecal samples” which discussed Bacillus subtilis as an important part of the gut microbiota to be used as a commonly used probiotic, the following comments should be followed:

-          All names of bacteria (genus and species) as well as all names of animals (genus and species) should be italic in whole part of the manuscript as well as in references section.

-          I have a special concern about statistics that they are missed in the methodology so to my opinion the results are not accurate, so please perform statistical analysis and rewrite results and discussion.

-          Ethical approval should be added.

-          The word strain/strains should be replaced by isolates in whole manuscript, that strains are already known and obtained from strain bank, but here you isolate Bacillus subtilis from your samples so they are isolates.

-          The words in vitro should be italic as in vitro in whole the manuscript even in references section.

-           Several words required spaces between them e.g. line no. 22, so please check the whole manuscript and correct.

-          Abstract: should be re-written to be more comprehensive about the highlighted result of biological characterization should be added in order to sounds better.

-          Introduction is too short to be representative about all parameters of this study, please rewrite.

-          Line 65: what is the indication of no. 3 after Bacillus subtilis.

-          Materials and Methods:

·         Add the references in each step because some of them are missed.

·         You did not clarify the diet of minks from which you collected the samples, please add.

·         Please add the name of media in which you collected samples, transported to lab as well as centrifuged the feces in them.

·         Line 83: 1ml B. subtilis MG-1: you should add 1 ml of what media in which grown B. subtilis.

·         How did you identify B. subtilis MG-1? You should mention in materials and methods section.

·         Line 154: change thalli to colony that thalli are for fungi not bacteria, as well as why you used this method which used for fungi DNA extraction as your work is on bacteria as many references extracted DNA from bacteria?

·         Control group should be highlighted in writing more.

-          Results

·         This section was written in good details as mentioned in materials and methods with the same sequence in writing to cover all points of the work.

·         Line 393: Don not begin with abbreviation, as well as in whole manuscript you should first mention the complete words then abbreviate.

-          Discussion:

·         This section should be rewritten in more organized pattern as well as more details of the discussion of the result should be highlighted with recent relevant references.

·         Don’t start with literatures (this should be in introduction section), you should start with your obtained results then discuss deeply with regard to your work.

·         Line 445-449: should be removed as you did not work on immunity.

-          Conclusion: is very short, the conclusion should be rewritten in more different style than the present conclusion to give power to the manuscript.

-          References:

·         Should be updated till 2023.

·         More relevant references about the aim of the study should be added.

·         Write the all authors, names not et al. in all references.

-          Section of abbreviation at the end of the manuscript should be added.

Comments on the Quality of English Language

Minor editing of English language required

Round 2

Reviewer 1 Report

Comments and Suggestions for Authors

The authors have thoroughly revised the manuscript of the article. I believe that the article can be accepted for publication in its present form.

Comments on the Quality of English Language

Minor editing of English language required

Reviewer 2 Report

Comments and Suggestions for Authors

Dear Authors

Thanks for your kind effort in correction of the revised manuscript  

Best wishes